# Vaginal Leptothrix: An Innocent Bystander?

**DOI:** 10.3390/microorganisms10081645

**Published:** 2022-08-15

**Authors:** Pedro Vieira-Baptista, Joana Lima-Silva, Mario Preti, Carlos Sousa, Fernanda Caiano, Colleen K. Stockdale, Jacob Bornstein

**Affiliations:** 1Hospital Lusíadas Porto, 4050-115 Porto, Portugal; 2Lower Genital Tract Unit, Centro Hospitalar de São João, 4200-319 Porto, Portugal; 3Department of Surgical Sciences, University of Torino, 10124 Turin, Italy; 4Unilabs, 4150-178 Porto, Portugal; 5Department of Obstetrics & Gynecology, University Iowa Healthcare, Iowa City, IA 52242, USA; 6Galilee Medical Center, and Azrieli Faculty of Medicine, Bar Ilan University, Safed 1311502, Israel

**Keywords:** leptothrix, lactobacillosis, vaginal flora, dysbiosis, candidiasis, wet mount microscopy

## Abstract

Leptothrix are long bacteria of rare occurrence; although these bacteria have been implicated in causing vaginal symptoms identical to candidiasis, studies on prevalence and effect on overall vaginal health are lacking. In this study, we evaluated data of women referred to a private clinic for treating vulvovaginal symptoms (n = 1847) and reassessed data of our previous and ongoing studies (n = 1773). The overall rate of leptothrix was 2.8% (102/3620), and the mean age of affected women was 38.8 ± 10.65 years (range 18−76). The majority of the women with leptothrix had normal vaginal flora (63.7% [65/102]). Leptothrix was associated with a higher risk of candidiasis (relative risk (RR) 1.90, 95% confidence interval (CI) 1.1600–3.1013; *p* = 0.010) and a lower risk of bacterial vaginosis (RR 0.55, 95% CI, 0.3221–0.9398; *p* = 0.029) and cytolytic vaginosis (RR 0.11, 95% CI, 0.0294–0.4643; *p* = 0.002). No cases of trichomoniasis were observed. Human immunodeficiency virus infection increased the risk of leptothrix (RR 3.0, 95% CI, 1.6335–5.7245; *p* = 0.000). Among the women evaluated for vulvovaginal symptoms, 2.4% (45/1847) had leptothrix, and in 26.7% (12/45), leptothrix was considered the causative entity. This study suggests that leptothrix occurrence is rare; it remains unresolved if it can be a cause of vulvar symptoms.

## 1. Introduction

The term “lactobacillosis” was first used by Horowitz et al. in 1994 and originally described as the presence of long forms of serpiginous lactobacilli (60 μm), usually referred to as “leptothrix,” from the Greek “λεπτος” (thin) and “ϑρίξ” (hair) (Figure 1). Horowitz described these forms of long bacteria in the vagina of women with symptoms similar to those of candidiasis (burning, itching, and irritation), typically starting 7–10 days after menses [1,2]. However, most women presenting with these forms of lactobacilli were asymptomatic [1]. Some authors classify an increased number of lactobacilli, in the absence of cytolysis, independently of their characteristics, as lactobacillosis as well. If a cytolytic pattern is concomitantly present, it is classified as cytolytic vaginosis, and it may be speculated that the two presentations may represent the same entity or consist of a part of a spectrum [3].

While “leptothrix” and “leptotrichia” have been used in the literature to describe different bacterial genera, in this paper, we refer to leptothrix as the long bacteria that sometimes are found in vaginal discharge and are usually considered a part of the *Lactobacillus* genus, but studies in this field are scarce [4,5]. To date, it has not been established whether it is a specific species or an ordinary lactobacillus species that acquires this morphology as a result of external factors such as exposure to antibiotics or antifungals, other microorganisms sharing the same niche, or certain nutritional factors [1,4,6]. Since antifungals or antibiotics are routinely employed to treat vaginal infections, the theory of pressure” caused by exposure to antifungals or antibiotics, a common event among bacteria, must be seriously considered [1,7].

Based on a limited number of previous studies, leptothrix prevalence ranges between 0.5–15.2% [2,8,9]. In one of the studies, leptothrix was also found to be simultaneously present with *Trichomonas vaginalis*, and thus, the authors recommended excluding the presence of this protozoan when leptothrix is found [1]. Furthermore, leptothrix was also associated with vulvar pain in one study; however, a clear distinction between leptothrix and cytolytic vaginosis was not established [6].

This study aimed to determine whether leptothrix is pathogenic or an innocent bystander. We defined the following main objectives: (1) to identify the rate of leptothrix occurrence among women who were referred to a private practice with symptoms of vulvovaginitis and determine the extent to which it was considered the cause of symptoms and (2) to assess the prevalence of leptothrix in various populations using the data of previous studies from our group, in which wet mount microscopy (WMM) was performed. Secondary objectives included examining the outcome of treatment, risk factors, changes in vaginal pH, and associated conditions.

## 2. Materials and Methods

We retrospectively evaluated the results of phase-contrast WMM samples collected from two groups of women to identify leptothrix cases. Group A included 1847 women who were referred to a private clinic due to vulvovaginal symptoms, dyspareunia, recurrent urinary tract infections, or persistent inflammatory Pap tests. 

Group B consisted of women from the following four studies of our group: (1) performed to validate a molecular test for the diagnosis of vaginitis [10], (2) evaluated the impact of vaginal flora on fertility treatments [11], (3) evaluated the vaginal flora by WMM in women living with human immunodeficiency virus (HIV) [12], and (4) an ongoing study, which evaluated the impact of methylation in a cervical cancer screening program and correlated WMM and cervical intraepithelial neoplasia. In all studies included in group B, pregnancy was an exclusion criterion.

One of the studies was designed to assess the presence of *T. vaginalis*, by using a validated molecular test. This study was also the only one in which the pH was systematically evaluated and recorded [9].

All samples were collected from the right lateral vaginal wall, using an endocervical brush, often without using a speculum; when a speculum was used, it was unmoistened. We opted to use endocervical brushes because it allows the preparation of an even sample without absorbing liquid or leaving fibers on the slide. Subsequently, the samples were spread onto a glass slide, air-dried, rehydrated, and read according to the International Society for the Study of Vulvovaginal Disease recommendations [3]. Immediate reading was not performed for convenience and blind reading [10]. The leptothrix diagnosis was based on the presence of long (>60 μm), nonbranching, filamentous lactobacilli, regardless of their amount and other concomitant findings [3]. All readings were performed by the same author, using a Leitz Biomed microscope (Wetzlar, Germany) equipped with phase contrast at 400× magnification.

Demographic data, including age and parity, smoking habits, comorbidities, signs and symptoms, culture results for *Candida* spp., polymerase chain reaction data for *T. vaginalis*, and the outcome of treatments prescribed, including antibiotics and vaginal irrigation, were retrieved from the electronic records of the institution. In group A, demographic data, signs, and symptoms were compared between patients whose symptoms were considered due to leptothrix and other causes. The symptoms were attributed to the presence of leptothrix when other conditions did not explain it.

Data analysis was performed using Microsoft Excel^®^ for Mac version 16.58 (Microsoft Corporation, Redmond, WA, USA) and SPSS version 26.0 (IBM, Armonk, NY, USA). The χ^2^ test and *t*-test were used for nominal and continuous variables, respectively. Continuous variables are presented as means and standard deviations, while categorical variables as percentages. When relevant, the relative risk (RR) was calculated, namely the risk in women living with HIV and associated candidiasis, bacterial vaginosis (BV), aerobic vaginitis/desquamative inflammatory vaginitis (AV/DIV), trichomoniasis, and cytolytic vaginosis. A value of *p* < 0.05 was considered statistically significant.

## 3. Results

Leptothrix was identified in 2.4% (45/1847) of group A (women with vulvovaginal symptoms); in 12/45 cases, leptothrix was considered the sole cause of symptoms, while in the other 33 cases, the most common causes were candidiasis (12, nine by *C. albicans* and three by non-*albicans* species), vulvodynia (eight), and lichen sclerosus (seven) (Table 1).

Table 1 shows the demographic characteristics of women with leptothrix divided according to the final diagnosis, leptothrix vs. other diagnoses despite the presence of leptothrix. No significant differences were observed in terms of age, smoking status, menopausal status, or contraception. The mean duration of the symptoms before diagnosis was significantly shorter in the leptothrix group than in the other diagnoses group. Further, more women in the leptothrix group noted abnormal discharge and had consumed significantly more antibiotics (100% [12/12]) or probiotics (25% [3/12]) in the six-month period preceding sample collection (Table 1). In women with leptothrix, 10/12 (83.3%) were referred due to vulvar burning (4 out of these 10 also reported itching), 1/12 complained of dyspareunia, and 1/12 had an increased discharge. Of note, when specifically questioned, dyspareunia was reported by seven women (Table 1). The gynecological examination was unremarkable, except in one case each of vulvar redness and vaginal enanthema. Eleven group A women received treatment; the one woman who had no other symptoms, but discharge was reassured with the diagnosis and not given treatment (Table 2).

Group B involved a total of 1773 women (Table 3). The global prevalence of leptothrix in this group was 3.2% (57/1773) and ranged from 2.3 to 8.1% among the four included studies. The highest prevalence was encountered in the study on women living with HIV. These women had a RR of 3.0, 95% confidence interval (CI) 1.6335 to 5.7245, *p* = 0.000. Excluding this study, the global prevalence dropped to 2.8% (47/1650). In most of the group B leptothrix cases, the WMM slides were classified as normal (68.4% [39/57]), followed by BV (14.0% [8/57]), presence of *Candida* spp. (10.5% [6/57]), AV/DIV (5.3% [3/57]), and cytolytic vaginosis (1.8% [1/57]). Further, in the study that was designed to test for *T. vaginalis* (using a molecular test), it was negative in all identified 22 cases of leptothrix. 

When the data of groups A and B were considered together, the rate of leptothrix was 2.8% [102/3620]), and the majority of these women had normal flora (63.7% [65/102]). Inflammation was uncommon (13.7% [14/102]), and the lactobacillary grades were mostly normal (grade I, 66.7% [68/102]; grade IIa, 8.8% [9/102]) (Table 3). Further, although the presence of concomitant leptothrix and *Candida* spp. was higher in group A than in group B, statistical significance was not observed (24.4% vs. 10.5%, *p* = 0.061). The mean age of the 102 patients identified was 38.8 ± 10.65 years (range 18−76 years) (Figure 2); 8.8% (9/102) were postmenopausal (Table 3). Smoking was reported in 17.6% of women.

We did not find an association between the presence of leptothrix and histological high-grade squamous intraepithelial lesions in women referred for colposcopy (11.1% [1/9] vs. 20.6% [79/383] if leptothrix was absent, *p* = 0.484). Further, a trend toward better pregnancy-related outcomes in women who received fertility treatments was observed; however, the difference was not statistically significant (Table 3). 

The vaginal pH, in the majority (77.3% [17/22]) of tested patients, was within the “normal” range (≤4.7) (Figure 3). Of the five cases with pH > 4.7, two had BV, and one had AV/DIV, while the other two were normal [10].

Comparing women with (n = 47) and without (n = 1489) leptothrix in three of the studies of group B, excluding the study on women living with HIV, those with leptothrix had a significantly lower detection rate of associated BV (RR 0.55, 95% CI, 0.3221–0.9398; *p* = 0.029) and cytolytic vaginosis (RR 0.11, 95% CI, 0.0294–0.4643; *p* = 0.002). At the same time, the risk for the presence of *Candida* spp. was significantly higher (RR 1.90, 95% CI, 1.1600–3.1013; *p* = 0.010). Further, no differences in AV/DIV and trichomoniasis (RR 1.1, 95% CI 0.3418–3.4063, *p* = 0.897 and RR 0.87, 95% CI 0.0538–13.9352, *p* = 0.919, respectively) was observed in patients with and without leptothrix.

## 4. Discussion

In this study, we detected a leptothrix prevalence rate of 2.8%. This rate remained constant in different settings, including symptomatic and asymptomatic women. However, our data showed that the prevalence was higher in women living with HIV. Further, the leptothrix occurrence was more common in premenopausal women than that in postmenopausal women, with a peak prevalence age of 31−40 years.

Leptothrix is most likely an innocent bystander. Even when it was identified in women with vulvovaginal symptoms, in about ¾ there was another explanation for the symptoms. It was considered a possible cause for the symptoms in only 0.65% [12/1847] of the cases; in these cases, vulvar burning, in addition to abnormal discharge, was the most common symptom. Of note, in symptomatic women in whom a diagnosis of leptothrix was assumed, the duration of the symptoms was shorter than in those who had an alternative diagnosis; we may speculate that the intensity of the symptoms drove them to seek specialized help more expeditiously. Although we could not confirm previous claims that leptothrix is associated with trichomoniasis, we observed a trend toward an association of leptothrix with the presence of *Candida* spp. [1]. In one of the evaluated settings, for which data on previous treatments were available, most women had been previously treated with antibiotics and/or antifungals.

The pH was within the normal range in most of the cases. In the group of women with increased pH, the cause was probably the presence of BV and/or AV/DIV. Despite the small sample size, we can assume that the vaginal pH in women with leptothrix is likely determined by the background flora. Accordingly, the presence of leptothrix was associated with a reduced risk of BV and cytolytic vaginosis, but an increased risk of candidiasis.

We believe that the molecular approaches currently used to study the vaginal microbiome may clarify the role of leptothrix. Further, molecular diagnoses need to be correlated with microscopic findings and symptoms.

Despite the association between antibiotic and/or probiotic use before the diagnosis of the presence of leptothrix, we could not establish a temporal relationship with its development. However, the fact that leptothrix is more common in women living with HIV raises the question of whether the use of antiretrovirals or the infection itself may encourage leptothrix development. The latter is less likely because many of these women had absent to low viral loads.

In this study, treatment-related limitations were nonstandardized treatment approaches, treatment data availability for only a few women, and loss of substantial numbers of women during follow-up. Hence, the apparent success of treatment in some cases cannot be considered evidence of effectiveness. A correlation between symptoms persistence/resolution and the presence/absence of leptothrix following treatment was not systematically recorded, weakening any possible conclusions. It can be argued that the presence of leptothrix might have been a self-limiting event, which resolved spontaneously independent of the treatment measures employed. The use of antibiotics in women with leptothrix should be considered in only a very small group of women, after considering other possible alternative diagnostics; its potential risks and benefits must be thoroughly discussed.

In another study by our group, conducted in Príncipe, São Tomé and Príncipe, we found a very low rate of leptothrix (0.3% [3/1141]). Interestingly, this population had a very high rate of abnormal vaginal flora (82.5%), with most women having BV (54.6%) followed by AV/DIV (25.8%). Therefore, we hypothesized that the presence of leptothrix might have significant ethnic and/or geographical differences, because of, or in association with, a lower risk of BV [13]. Hence, leptothrix is a marker of eubiosis, or of a recovering microbiome, rather than one of dysbiosis—or a mere innocent bystander. Consistent with this idea, we could not establish an association of leptothrix with negative outcomes in terms of cervical intraepithelial lesions or fertility treatments.

Of note, most women were referred or sought specific help due to long-standing and/or bothersome symptoms; in most cases, the diagnosis was not previously made, highlighting the need for more medical education in the field of vulvovaginal diseases, including the use of WMM.

The strengths of this study include the high number of women evaluated, WMM data analyzed by an experienced author, and use of a systematic approach. The study was limited by the unavailability of pH determination in the majority of cases, and only one study used a molecular test to evaluate the presence of *T. vaginalis*. In addition, the number of enrolled pregnant women was low, and data on time since delivery and/or breastfeeding are missing

## 5. Conclusions

Leptothrix is an uncommon finding, and it remains unclear if it ever causes vulvar symptoms. It does not seem to be associated with negative outcomes but was more common in women living with HIV. Leptothrix is not a marker of dysbiosis.

## Figures and Tables

**Figure 1 microorganisms-10-01645-f001:**
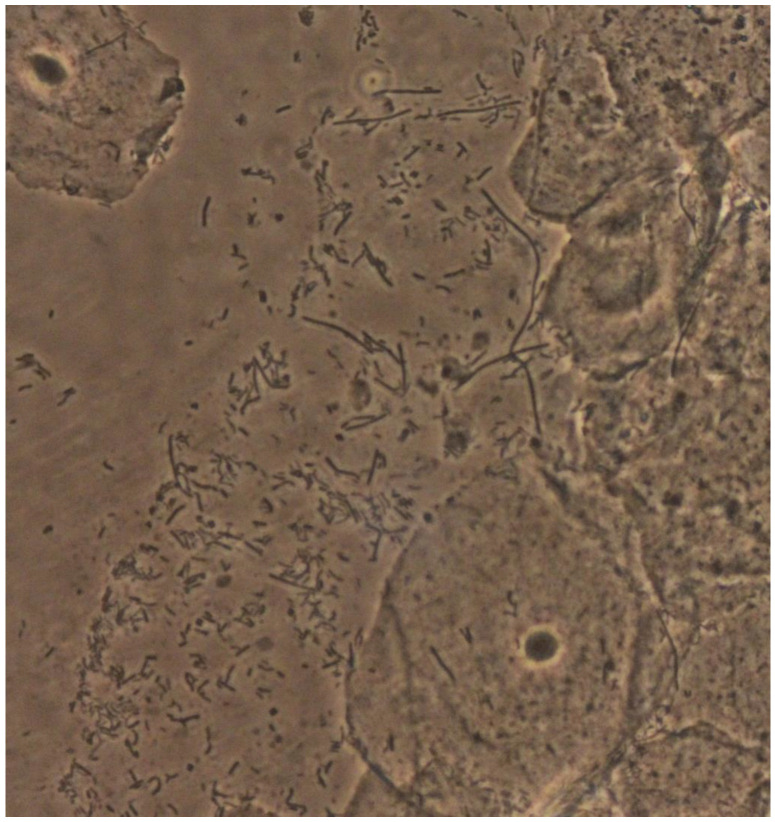
Leptothrix and normal background flora (×400).

**Figure 2 microorganisms-10-01645-f002:**
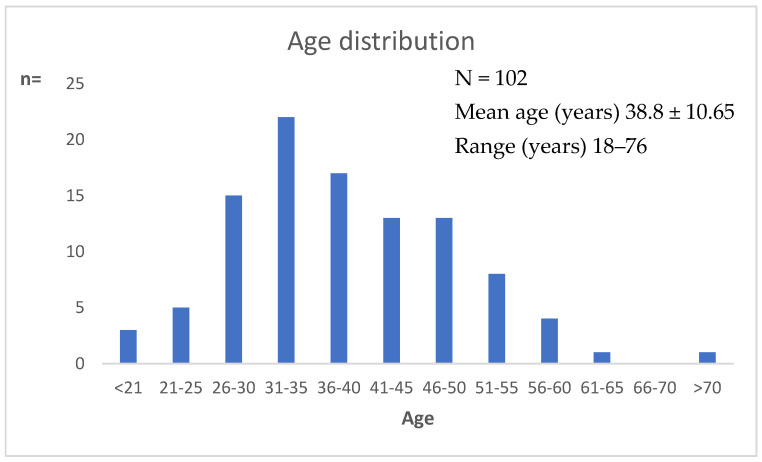
Age distribution of women with leptothrix.

**Figure 3 microorganisms-10-01645-f003:**
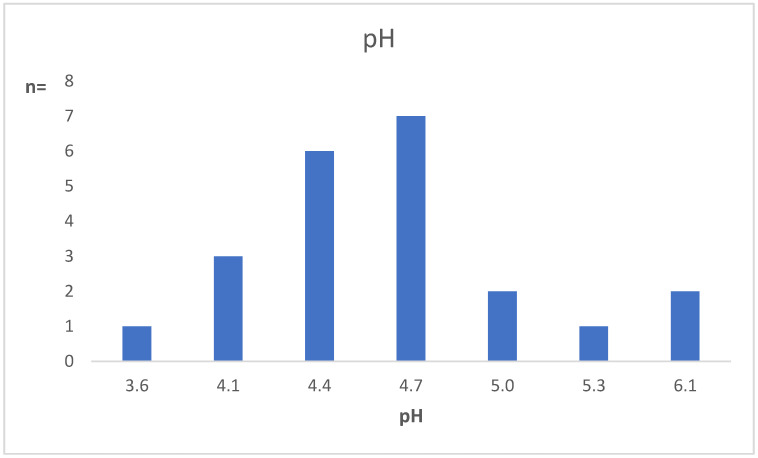
Vaginal pH distribution in women with leptothrix.

**Table 1 microorganisms-10-01645-t001:** Characteristics of women referred for an appointment due to vulvovaginitis and in which leptothrix was found. Women were divided into 2 groups according to whether leptothrix was considered as the final sole diagnosis or if another diagnosis could explain the symptoms.

	Leptothrix	Other Diagnosis	*p*=
Age (mean ± SD) (years)	35.5 ± 9.27	37.4 ± 11.27	0.604
Smoking	8.3% (1/12)	6.1% (2/33)	0.787
Menopause	0% (0/12)	6.1% (2/33)	0.383
Duration of symptoms (mean ± SD) and range (months)	12.8 ± 9.36 (2–27)	54.5 ± 67.57 (1–240)	0.040
Contraception			
- None	25.0% (3/12)	38.7% (12/31)	0.397
- Condom	0% (0/12)	3.2% (1/31)	0.529
- Combined oral contraceptives	50.0% (6/12)	41.9% (13/31)	0.633
- Progestin only pill	16.7% (2/12)	3.2% (1/31)	0.121
- Intrauterine devices	8.3% (1/12)	9.7% (3/31)	0.890
- Vaginal ring	0% (0/12)	3.2% (1/31)	0.529
Symptoms			
- Burning	91.7% (11/12)	63.6% (21/33)	0.067
- Itching	33.3% (4/12)	45.4% (15/33)	0.467
- Dryness	0% (0/12)	12.1% (4/33)	0.206
- Pain (spontaneous)	16.7% (2/12)	15.2% (5/33)	0.902
- Dyspareunia	70% (7/10)	67.8% (19/28)	0.899
- Dysuria	8.3% (1/12)	12.1% (4/33)	0.720
- Discharge	66.7% (8/12)	33.3% (11/33)	0.045
- Cyclical symptoms	58.3% (7/12)	41.9% (13/31)	0.334
Previous treatments			
- Antifungals	75.0% (9/12)	60.1% (20/33)	0.372
- Antibiotics	100% (12/12)	24.2% 8/33	0.000
- Probiotics	25.0% (3/12)	3.0% (1/33)	0.022
Final diagnosis			
- Aerobic vaginitis/desquamative inflammatory vaginitis		1	
- Bacterial vaginosis		1	
- Candidiasis		12	
- Cytolytic vaginosis		1	
- *Granuloma fissuratum*		1	
- Lichen sclerosus		7	
- Pelvic floor dysfunction		1	
- Vulvodynia		8	
- Vulvar involvement of Crohn’s disease		1	

SD: standard deviation.

**Table 2 microorganisms-10-01645-t002:** Age, reason for referral and additional symptoms, treatments prescribed, and outcome of women in group A with a final diagnosis of leptothrix. ^1^—metronidazole 500 mg vaginal ovules *id* for 10 days and then 2–3 times a week for 3 months; ^2^—amoxicillin 875 mg + clavulanate 125 mg pills, twice a day for 10 days; ^3^—sodium bicarbonate 30–40 g/L used in vaginal irrigations twice a day for 2 weeks and then as needed for symptom control; ^4^—dequalinium chloride 10 mg vaginal pills once a day for 12 days and then 2–3 times a week for 3 months.

Case	Age	Reason of Referral	Other Symptoms	Treatment	Outcome	Notes
1	44	Burning and itching	Dyspareunia and discharge	Metronidazole ^1^	Lost to follow-up	
2	25	Dyspareunia	Burning and discharge	Metronidazole ^1^	Lost to follow-up	
3	29	Burning and itching	Discharge	Metronidazole ^1^	Partial improvement	No further improvement with sodium bicarbonate
4	51	Burning	Dyspareunia and discharge	Metronidazole ^1^	Lost to follow-up	
5	29	Burning	Dyspareunia and discharge	Amoxicillin + clavulanate ^2^	No improvement	Asymptomatic with metronidazole
6	45	Discharge	None	None	Not applicable	
7	41	Burning	Burning	Sodium bicarbonate ^3^	Asymptomatic	
8	24	Burning	Dyspareunia	Dequalinium chloride ^4^	Asymptomatic	
9	32	Burning	Discharge	Sodium bicarbonate ^3^	Partial improvement	No further improvement with sodium bicarbonate or amoxicillin + clavulanate
10	31	Burning	Discharge	Metronidazole ^1^	Asymptomatic	
11	46	Burning and itching	Dyspareunia	Sodium bicarbonate ^3^	Lost to follow-up	
12	29	Burning and itching	Dyspareunia and discharge	Metronidazole ^1^	Partial improvement	

**Table 3 microorganisms-10-01645-t003:** Evaluation of previous and ongoing studies by the authors in which wet mount microscopy was performed and data about leptothrix is available on file. The total number of cases may differ from the figures shown in the published studies, as we evaluated all cases available, regardless of inclusion in the final analysis. Inflammation was defined as >10 leukocytes per high power field.

Group	Study	Year of the Study	Objectives	Population	N	Rate of Leptothrix	Associated Conditions	Comments
A	Evaluation of women with vulvovaginal symptoms	2022	To evaluate the prevalence and characteristics of women with leptothrix	Women attending a private practice with vulvovovaginal symptoms (16–78 years old)	1847	2.4% (45/1847)	BV = 13.3% (6/45) AV/DIV = 2.2% (1/45) CV = 2.2% (1/45) *Candida* spp. = 24.4% (11/45) Trichomoniasis = 0% (0/4) § Inflammation = 6.7% (3/45) Lactobacillary grades: - I = 71.1% (32/45) - Iia = 11.1% (5/45) - Iib = 15.6% (7/45) - III = 2.2% (1/45)	The diagnosis of leptothrix was considered in 26.7% (12/45) of cases with leptothrix
B	Ongoing (study evaluating methylation markers and vaginal flora in HPV positive women)	2022	To evaluate the correlation between cervical dysplasia, HPV, cervical dysplasia, methylation markers and vaginal flora	Women referred for colposcopy from the organized screening program (25–65 years of age)	392	2.3% (9/392)	BV= 22.2% (2/9) AV/DIV = 0.0% (0/9) CV = 0.0% (0/9) *Candida* spp. = 0.0% (0/9) Trichomoniasis = NA Inflammation = 0.0% (0/9) Lactobacillary grades: - I = 55.6% (5/9) - Iia = 22.2% (2/9) - Iib = 22.2% (2/9) - III = 0.0% (0/9)	One case of CIN3/AIS There was no statistically significant difference in the rate of CIN2+ between women with and without leptothrix (11.1% [1/9] and 20.6% [79/383], *p* = 0.484)
	Wet Mount Microscopy of the Vaginal Milieu Does Not Predict the Outcome of Fertility Treatments: A Cross-sectional Study [11]	2022	To evaluate if women with dysbiosis subjected to fertility treatments have lower rates of pregnancy.	Women who underwent fertility treatments at a fertility clinic (22–40 years of age)	500	3.2% (16/500)	BV = 6.2% (1/16) AV/DIV = 0.0% (0/16) CV = 0.0% (0/16) *Candida* spp. = 12.5% (2/16) Trichomoniasis = NA Inflammation = 6.2% (1/16) Lactobacillary grades: - I = 56.2% (9/16) - Iia = 25% (4/16) - Iib = 12.5% (2/16) - III = 6.2% (1/16)	There were no differences according to the presence or absence of leptothrix in biochemical pregnancy (28.6% vs. 27.5%, *p* = 0.931), clinical pregnancy (7.1% vs. 23.2%, *p* = 0.157) or delivery of a live fetus (7.1% vs. 15.4%, *p* = 0.398)
	Clinical validation of a new molecular test (Seegene Allplex™ Vaginitis) for the diagnosis of vaginitis: a cross-sectional study [10]	2021	To validate a molecular assay in the diagnosis of candidiasis, BV and trichomoniasis.	Consecutive symptomatic and asymptomatic women (18–60 years of age)	758	2.9% (22/758)	BV = 13.6% (3/22) AV/DIV = 9.0% (2/22) CV = 4.5% (1/22) * *Candida* spp. = 9.0% (2/22) Trichomoniasis = 0.0% 0/22) Inflammation = 6.2% (1/16) Lactobacillary grades: - I = 81.8% (18/22) - Iia = 0.0% (0/22) - Iib = 13.6% (3/22) - III = 4.5% (1/22)	
	Wet mount characterization of the vaginal flora of HIV women [12]	2017	To evaluate the vaginal flora in women living with HIV	Women living with HIV (18–76 years of age	123	8.1% (10/123)	BV = 20% (2/10) AV/DIV = 10.0% (1/10) CV = 0% (0/10) *Candida* spp. = 20% (2/10) Trichomoniasis = NA Inflammation = 0% (0/10) Lactobacillary grades: - I = 40% (4/10) - Iia = 0% (0/10) - Iib = 130% (3/10) - III = 30% (3/10)	All women with leptothrix were using antiretroviral treatment Only one was symptomatic (burning); the background flora was normal
Total		-	-	-	3620	2.8% (102/3620)	BV = 13.7% (14/102) AV/DIV = 3.9% (4/102) CV = 2.0% (2/102) *Candida* spp. = 16.7% (17/102) Trichomoniasis = 0% (0/26) Inflammation = 13.7% (14/102) Lactobacillary grades: - I = 66.7% (68/102) - Iia = 8.8% (9/102) - Iib = 16.7/% (17/192) - III = 5.9% (6/102)	

* Agreement between culture, PCR and WMM; § only cases in which PCR was performed. AIS: adenocarcinoma in situ; BV: bacterial vaginosis; AV/DIV: aerobic vaginitis/desquamative inflammatory vaginitis; CIN: cervical intraepithelial neoplasia; CV: cytolytic vaginosis; NA: not available.

## Data Availability

The data presented in this study are available on request from the corresponding author. The data are not publicly available due to privacy issues.

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
