# Peer review of "Vaginal Leptothrix: An Innocent Bystander?"

_microorganisms, 2022, doi:10.3390/microorganisms10081645_

Round 1
Reviewer 1 Report
The authors, all highly respected members of the academic community dealing with vulvovaginal disease, embark upon the task, or journey, to establish whether Leptothrix species or morphotype observed during wet mount microscopy are indeed vaginal pathogens capable of causing lower genital tract symptoms and worthy of treatment. Not an easy task, and resembles a similar task validating the entity of cytolytic vaginosis to establish whether Leptothrix species are pathogenic and capable of causing symptoms is difficult and has profound consequences. If readers of the manuscript come away convinced that leptothrix may cause vaginal burning, a controversial unvalidated precedent is established requiring the “innocent bystander” practitioner to treat these microorganisms using unidentified therapeutic agents of unproven efficacy and with numerous potential adverse effects known to be related to antibiotics. The responsibility is huge!
How does one establish causality to the not-uncommon vaginal lactobacillus spp. since they are also observed in healthy asymptomatic women? This is a crucial question and whether the authors, in spite of the large number of specimens evaluated from a variety of studies, answer this question is unclear. The overall prevalence of Leptothrix in healthy matched control women is not established, but that the prevalence is extremely low in the study population in whom specimens were available.
Even when identified in group A, almost ¾ of the 45 symptomatic women had another possible or likely cause of symptoms.
The study is heavily over analyzed and somewhat confusing for the reader. In reading the post discussion conclusion, the points made are quite clear and not controversial. Indeed, Leptothrix is an uncommon finding and which very rarely may be associated with burning (? If ever). No association with negative outcome was established by the analysis performed. This in its own right makes the manuscript publishable. Yes the final line in the abstract creates the potential for controversy and use of unknown but empiric therapy viz……”but the microorganism can be associated with vulvar burning”….a contradiction of everything else in the manuscript…..needs deletion?
Clearly the manuscript ignores Koch principle of causation. This is also not a systematic approach to proving the pathogenicity of Leptothrix using comparison groups and indicating that symptoms resolve when Leptothrix eradicated or symptoms persist when Leptothrix survives/persists.
Author Response
Dear editor and reviewers,
The authors would like to thank for your thoughtful and wise comments and corrections, which certainly will contribute to the improvement of the manuscript.
We made the requested changes in the manuscript and present here an itemized answer/comment to all issues raised.
We hope we managed to fulfil all the requests.
Please see the attachment

Reviewer 2 Report
Vieira-Baptista and colleagues have an elegant setup combining previously published and novel data to estimate the pathogenic potential of so-called leptothrix observed in vaginal wet mounts.
Confusingly, both the names Leptothrix and Leptotrichia have been independently assigned to other bacterial genera, most likely not related to the filaments observed. The authors do a good job explaining that they are describing a morphological finding, but this could be made even more explicit in the first paragraph of the introduction.
An interesting analysis that could be added, if data is available, is whether any relationship exists between recent pregnancy/vaginal delivery/breastfeeding and detection of leptothrix. This could partially explain the higher prevalence in the 31-40 age bracket. Other recent studies have found a relationship between parity and increased prevalence of pre-menopause vaginal dysbiosis.
Author Response

(The authors gave the same response as above.)

Round 2
Reviewer 1 Report
I have reviewed the corrected/revised manuscript and NOW formally recommend ....ACCEPT with no need for further revision